# Biphasic Functions of Sodium Fluoride (NaF) in Soft and in Hard Periodontal Tissues

**DOI:** 10.3390/ijms23020962

**Published:** 2022-01-16

**Authors:** Xingzhi Wang, Nitesh Tewari, Fuyuki Sato, Keiji Tanimoto, Lakshmi Thangavelu, Makoto Makishima, Ujjal K. Bhawal

**Affiliations:** 1Department of Biochemistry, Nihon University School of Medicine, Tokyo 173-8610, Japan; wangxingzhi1234@outlook.com; 2Centre for Dental Education and Research, Division of Pedodontics and Preventive Dentistry, All India Institute of Medical Sciences, New Delhi 110029, India; dr.nitesht@gmail.com; 3Shizuoka Cancer Center, Pathology Division, Shizuoka 411-8777, Japan; fsatodec1dec2@yahoo.co.jp; 4Department of Translational Cancer Research, Research Institute for Radiation Biology and Medicine, Hiroshima University, Hiroshima 734-8553, Japan; ktanimo@hiroshima-u.ac.jp; 5Center for Transdisciplinary Research, Department of Pharmacology, Saveetha Dental College, Saveetha Institute of Medical and Technical Sciences, Chennai 600077, India; lakshmi@saveetha.com; 6Department of Biochemistry and Molecular Biology, Nihon University School of Dentistry at Matsudo, Chiba 271-8587, Japan

**Keywords:** fluoride, epithelial cells, mesenchymal stem cells, osteoblasts, osteoclasts, periodontal disease, miRNA

## Abstract

Sodium fluoride (NaF) is widely used in clinical dentistry. However, the administration of high or low concentrations of NaF has various functions in different tissues. Understanding the mechanisms of the different effects of NaF will help to optimize its use in clinical applications. Studies of NaF and epithelial cells, osteoblasts, osteoclasts, and periodontal cells have suggested the significant roles of fluoride treatment. In this review, we summarize recent studies on the biphasic functions of NaF that are related to both soft and hard periodontal tissues, multiple diseases, and clinical dentistry.

## 1. Introduction

Fluoride is well known for its use in the treatment of dental caries, either systemically or topically [1]. Fluoride intake (such as in drinking water, fluoridated toothpaste, or fluoride supplements) is the cornerstone to preventing dental caries in adults and children. Fluoride prevents dental caries by slowing down the demineralization of enamel, which is caused by the interaction between dental plaque and dental hard tissues. Fluoride may inhibit tooth decay by 40–60% by co-precipitating calcium and phosphate ions and by enhancing the precipitation of fluoridated apatite. Fluoride is also found deposited as calcium fluoride in dental plaque, which helps to prevent dental caries [2]. However, a high concentration of fluoride alters the mineralization process and can cause hypomineralization or fluorosis [3]. It has been demonstrated that a concentration of fluoride between 1.6 and 1.8 ppm in the drinking water is the risk threshold for dental fluorosis [4]. Osteoporosis and osteosclerosis are induced by high concentrations of sodium fluoride (NaF) in the drinking water [5]. Genotoxicity has been found to be related to oxidative stress or cell cycle disturbances induced by high concentrations of fluoride [6].

The effects of fluoride are biphasic. Exposing human dental pulp cells to a low concentration of NaF (25 μM–50 μM) stimulated proliferation and differentiation, while a concentration of NaF at 50 μM to 100 μM stimulated proliferation, differentiation, and collagen synthesis in human osteoblastic osteosarcoma cells [7]. Myoblasts exposed to low concentrations of NaF promoted proliferation, which could be regarded as a muscle enhancing factor, and high concentrations elevated the expression of muscle atrophy-related genes, myostatin and atrogin-1, reactive oxygen species (ROS), and inflammatory factors that accelerate skeletal muscle atrophy [8]. Fluoride can alter the bone mineral metabolism and can affect the homeostasis of bone formation and resorption [9]. Early studies demonstrated that the intake of fluoridated water can inhibit hydrocortisone-induced bone resorption and parathormone-induced alveolar bone resorption [10,11]. Using fluoride as a therapeutic agent to treat postmenopausal osteoporosis in adults has also been investigated [12].

High and low concentrations of fluoride result in toxic or tolerated dosages, respectively, making it vital to determine the concentration of fluoride intake. Various signaling pathways can alter cellular or metabolic actions in response to either high or low concentrations of fluoride. This review focuses on the various effects induced by high and low concentrations of fluoride.

## 2. Fluoride and Epithelial Cells

Epithelial cells function as a barrier to protect organisms from physical, chemical, and microbial damage and to maintain mucosal integrity [13]. Dental epithelial cells can differentiate into various types of cells that can secrete enamel matrix proteins during tooth development [14]. The matrix metalloproteinase-20 (MMP-20) is produced by ameloblasts and is responsible for cleaving other enamel matrix proteins such as amelogenin, ameloblastin, and enamelin [15]. MMP-20 is produced at the secretory stage of dental enamel formation and cleaves proteins that allow the crystals to grow into the space [15]. Ameloblasts also secrete kallikrein-4 (KLK4), which is essential for the breakdown of enamel proteins and enamel crystallization [16]. KLK4 helps crystals to grow and contract, which strengthens the enamel at the last stage of amelogenesis [16].

Recent studies have revealed that different concentrations of fluoride have various effects on epithelial cells.

### 2.1. High Concentration Fluoride and Epithelial Cells

A high concentration of NaF (millimolar level) induces endoplasmic reticulum stress, apoptosis, and DNA fragmentation and interferes with enamel proteinases [17]. Exposure to a high concentration of fluoride results in enamel hypomineralization, also known as enamel fluorosis [18]. The severity of enamel fluorosis increases depending on the volume of fluoride intake and the duration of fluoride exposure [18]. Fluoride decreases the abilities of proteinases such as MMP-20 and KLK4 that degrade enamel matrix proteins, and as a result, inhibits crystal growth [19]. Alternatively, the intake of fluoride lowers the pH, which inhibits the synthesis and secretion of KLK4 [20]. One study reported that excessive fluoride results in hypomineralization through the reduced expression of Forkhead box O1 (FOXO1) in dental epithelial cells [21]. FOXO1 is reduced after exposure to excessive fluoride, which might affect the expressions of KLK4 and amelotin (AMTN), which ultimately results in enamel fluorosis. One mM fluoride was found to be cytotoxic and induced apoptosis in human gingival epithelial cells. Treatment with 2 mM NaF disturbed the gene network accompanied by endoplasmic reticulum stress [22].

### 2.2. Low Concentration Fluoride and Epithelial Cells

Studies have shown that fluoride has biphasic effects on epithelial ameloblast-lineage cells at different concentrations. Reduced proliferation was observed at a fluoride concentration higher than 1 mM; however, lower concentrations of fluoride, e.g., around 16 μM, promoted the proliferation of epithelial ameloblast-lineage cells [23].

During gingival wound healing, human epithelial cells cover the wound, a process termed re-epithelialization. The process of wound healing includes epithelial cell proliferation and migration as well as the synthesis and deposition of extracellular matrix components [24]. Extracellular matrix proteins such as fibronectin and laminin-5 are expressed by migrating keratinocytes during wound healing. To understand the characteristics of gingival epithelial cells responding to fluoride, human primary epithelial cells were treated with NaF to characterize their effects on cellular physiology. Cell proliferation peaked at a concentration of 50 μM of NaF, and a higher concentration of NaF (at the millimolar level) reduced proliferation. Cells treated with 50 μM of NaF showed significantly more motility than non-treated cells in vitro. qRT-PCR analyses showed increased mRNA expression levels of fibronectin and laminin-5 in cells treated with 50 μM of NaF [25].

### 2.3. Fluoride and Epithelial–Mesenchymal Interactions

Epithelial–mesenchymal interactions (EMT) are important for the development of ectodermal organs and for tissue regeneration and wound healing, including cellular events such as cell adhesion, proliferation, differentiation, and maturation [26]. Homeostasis, inflammation, migration and proliferation, and remodeling are the four major processes of wound healing [27]. Fibroblast growth factor 2 (FGF2) and its receptor, fibroblast growth factor receptor 2 (FGFR2), are crucial for proliferation, migration, and protease production in epithelial cells [28]. FGF2 is associated with EMT by inducing mesenchymal characteristics in epithelial cells [29]. Fibroblast growth factor 7 (FGF7) is essential for epithelial morphogenesis [30]. Twist family BHLH transcription factor 1 (Twist1) is a critical mediator for EMT [31]. One study demonstrated that treatment with 50 μM of NaF induces the expression of FGF2 and FGF7, and Twist1 is also upregulated in vivo. It was also found that the untreated group had more persistent wounds in mice [32].

## 3. Fluoride and Bone Marrow Mesenchymal Stem Cells (BMMSCs)

BMMSCs present in the bone marrow are self-renewing precursor cells that have the multipotency to differentiate into osteoblasts, chondroblasts, adipoblasts, and stromal cells [33]. Upon bone injuries, BMMSCs differentiate into osteoblasts and release growth factors during wound healing [34]. Consistent with other studies, fluoride also has dual effects on BMMSCs. A low concentration of NaF (50 or 500 μM) enhanced the proliferation of BMMSCs, while a high concentration of NaF (5 mM) reduced their proliferation. Furthermore, BMMSCs showed elevated motility and migration when compared with the control group. Treatment with 50 μM of NaF upregulated the expression of fibronectin and vimentin, which induced the Runt-related transcription factor (Runx2), induced osteoblast differentiation, and increased the secretion of osteocalcin (OCN) [35]. Treatment with a higher concentration of NaF induced cytotoxicity, DNA damage, and oxidative stress in BMMSCs [36]. Studies have indicated that fluoride-induced cytotoxicity depends on the concentration and the duration of fluoride exposure [37]. High concentrations of NaF (2 mM and above) result in oxidative stress and decreased viability and proliferation via the phosphor-c-Jun N-terminal kinase (JNK) pathway. In dental pulp stem cells, a low concentration of NaF promotes osteo/odontogenic differentiation via the PI3K/Akt pathway [38]. In addition, a low concentration of NAF (0.5 mM) is the optimal concentration to regulate the osteo/odontogenic differentiation of stem cells from apical papilla [39]. When exposed to mouse embryonic stem cells, a high concentration of NaF (over 1 mM) induced ROS and reduced DNA synthesis, which resulted in apoptosis through a JNK-dependent pathway [40].

## 4. Fluoride and Bone Metabolism

Bone homeostasis is maintained through bone formation by osteoblasts and bone resorption by osteoclasts [41]. The activity of osteoblasts and osteoclasts is critical for bone maintenance and remodeling. The process is regulated by many factors, such as sex hormones, parathyroid hormones, and calcitonin, as well as various growth factors and cytokines [42]. Osteoblasts and osteoclasts also can control each other’s formation, differentiation, apoptosis through multiple pathways via cytokines, extracellular proteins, and transcription factors [43]. Bone diseases such as osteoporosis are caused by dysfunctional bone remodeling or the disruption of the homeostasis maintained by osteoblasts and osteoclasts [43].

Both positive and negative effects of fluoride on bones and teeth have been well established [44,45]. Fluoride treatment increases the proliferation of osteoblasts and inhibits the function of osteoclasts [46]. Trace amounts of fluoride have been used to treat osteoporosis, vertebral fractures, and bone loss in patients [47], whereas excessive fluoride results in skeletal fluorosis [48]. Long-term exposure to fluoride can result in skeletal and dental fluorosis [49]. Thus, a better understanding of the effects of different concentrations of NaF on bone homeostasis will help to gain more insights into the usage of fluoride.

### 4.1. Fluoride and Osteoblasts

OCN and Osteopontin (OPN) are non-collagenous proteins secreted by osteoblasts that serve as markers for osteoblast maturation [50]. OCN promotes bone formation and regulates mineralization in the bone matrix, which has a complex regulatory network [51]. OPN is an extracellular matrix protein that has multiple functions associated with bone structuring and destruction in osseous tissues [52]. The transcription factor Runx2 induces the expression of osteoblastic genes including OCN and OPN [53]. Runx2 is expressed at different stages of osteoblasts (pre-osteoblasts, and immature and mature osteoblasts) and is essential for bone formation and osteoblastic differentiation [54]. Osterix (OSX), a zinc-finger containing a transcription factor, is also a downstream target of Runx2 [55]. Together with Runx2, OSX regulates the differentiation of pre-osteoblasts into mature osteoblasts and osteocytes, and the roles of OSX in maintaining bone homeostasis have been well studied [37]. One study showed that NaF treatment affects calcium homeostasis and transcription factor expression [56]. A low concentration of NaF induced Runx2 and OSX while a high concentration of NaF reduced their expression both at the mRNA and protein levels. Treatment of MC3T3-E1 cells with 50 or 500 μM of NaF enhanced their proliferation, alkaline phosphatase (ALP) activity, and extracellular matrix mineralization, as well as their expression of bone-related genes (Runx2, OSX, OCN, and OPN) [57]. Different concentrations of NaF were also found to differentially affect the expression of bone mineralized regulator proteins, such as OCN, OPN, osteonectin (ON), and bone sialoprotein (BSP) in bone marrow stromal cells. A higher concentration of NaF (10^−5^ M) decreased the level of the proteins while a lower concentration (10^−7^ M) increased them [44].

Various cellular mechanisms, including the Mitogen-activated protein kinase (MAPK) pathway, are proposed to function in bone formation during fluoride treatment [58]. It is well established that a low concentration of NaF (micromolar levels) regulates tyrosine kinase and ALP activity to increase osteoblast proliferation [59,60]. Fluoride treatment of osteoblasts is also biphasic in that low concentrations promote proliferation, whereas higher concentrations result in signs of weakened osteoblast activity [61]. Long periods of fluoride exposure reduce the expression of bcl-2 family proteins, which promotes apoptosis in osteoblastic cells [62]. Fluoride induces mitochondrial respiratory chain complex abnormal expressions, which in turn cause oxidative stress and result in apoptosis [63]. High concentrations of NaF also regulate the proliferation and cell cycle of the p16 gene methylation during the development of skeletal fluorosis [64]. The schematic diagram shown in Figure 1 summarizes the functions of NaF in osteoblasts.

### 4.2. Fluoride and Osteoclasts

Osteoprotegerin (OPG) is a soluble cytokine receptor of the tumor necrosis factor (TNF) receptor family that binds to the OPG ligand [65]. The receptor activator of nuclear factor kappa-B ligand (RANKL) is also a member of the TNF superfamily, which binds to RANK on target cells. OPG/RANKL/RANK is a key signaling pathway in bone metabolism. The overexpression of OPG alters osteoclast differentiation, and recombinant OPG impedes ovariectomy-induced bone loss in rats [66]. Recombinant OPG binds to OPG ligand on BMMSCs, thereby inhibiting osteoclast differentiation [67]. RANKL is an essential cytokine that regulates osteoclastogenesis and bone resorption [68]. In response to RANKL activation, the nuclear factor of activated T cells 1 (NFATc1) regulates the terminal differentiation of osteoclasts, and NFATc1-deficient embryonic stem cells were unable to differentiate into osteoclasts [69]. NFATc1 participates in the transcription of the genes involved in heart/valve septum formation, angiogenesis, T cell proliferation, and osteoclast formation. NFATc1 binds to the promoter of genes associated with bone resorption, including cathepsin K and matrix metalloproteinase 9 (MMP-9), thus inducing their gene expression [70].

Low concentrations of NaF (micromolar level) have little influence on the viability of BMMSCs and significantly downregulate both mRNA and the protein expression levels of NFATc1 in rat osteoclasts, which result in a reduction in the levels of cathepsin K and the attenuation of bone destruction [71]. Treatment with 0.5 mM to 1 mM of NaF inhibits the activity of osteoclasts in vitro [72]. Studies suggest that fluoride can act on matrix proteinases such as Metalloproteinases-2 and -9, thus inhibiting the matrix degradation [73,74]. Low concentrations of fluoride can possibly regulate osteoclasts via the B lymphocyte-induced maturation protein-1 (Blimp1)/B cell lymphoma 6 (Bcl6) axis which is a critical signaling pathway that regulates osteoclast differentiation and bone homeostasis [75]. On the other hand, the stimulation of NaF exhibits a U-shaped curve in a dose-dependent manner [76]. High concentrations of NaF induce RANKL and decrease OPG, thus increasing osteocyte-driven osteoclastogenesis via the RANK-JNK-NFATc1 signaling pathway [77]. A recent study found that osteoclasts demonstrate the most sensitivity to high concentrations of NaF with respect to other bone cell types and that fluoride exposure induces apoptosis via the Transforming growth factor (TGF)-β signaling pathway [78]. A schematic diagram of the functions of NaF in osteoclasts is shown in Figure 2.

## 5. Fluoride and Periodontal Diseases

Periodontal diseases are induced by bacterial biofilms caused by Gram-negative anaerobic bacteria [79]. Dental plaque formed after bacteria colonize teeth stimulates the host’s inflammation responses in gingival connective tissues [80,81]. Chronic periodontitis is mediated by interactions between the host and pathogens, which result in the apical migration of epithelial attachment and ultimately the destruction of connective tissue and alveolar bone [82,83].

A low concentration of NaF had anti-inflammatory effects that decreased the expression of IL-1β, IL-8, and TNF-α in human gingival fibroblasts [84]. A recent study demonstrated that NaF has an antibacterial activity against *P. gingivalis* in a concentration-dependent manner. A model of rat periodontitis treated with 500 μM of NaF in drinking water showed attenuated alveolar bone resorption and reduced levels of interleukins-6 and -8. The expression of NFATc1 was also reduced in osteoclasts and cathepsin K [71]. Another study found that treatment with NaF could reduce *P. gingivalis*-induced inflammation and bone loss in rat periodontitis tissues and significantly reduce the infiltration of polymorphonuclear leukocytes (PMN) [57].

Diabetes mellitus (DM) is associated with a reduction in insulin production or relative changes that increase glucose levels in the blood during insulin activity [85]. Hyperglycemia induced by increased glucose affects many tissues and organs, such as the kidneys, nerves, blood vessels, and periodontal tissues. Periodontitis is the sixth complication of DM, and they share various pathogenic mechanisms; however, the exact mechanism by which diabetes is associated with periodontitis is not fully understood [86]. One study reported that periodontitis models with diabetes suffer more alveolar bone loss than those without diabetes [87].

Insulin-like growth factors (IGFs) are involved in the development and growth of β cells that are important for β cells’ mitogenic action and inhibit β cell apoptosis [88]. The fetal and adult pancreas can produce IGFs and their binding receptors. The overexpression of IGF-2 causes apoptosis in islets and islet hyperplasia, while increased β cell mass elevates IGF-1 expression. The elevation of IGF-1 or mediators in the IFG signaling pathway protects β cells from apoptosis, thus preventing diabetes in mice [89,90,91].

IGFs are autocrine/paracrine factors that regulate the proliferation, differentiation and functions of osteoblasts, osteocytes, and osteoclasts [92,93]. Fluoride is possibly associated with IFG-1 to enhance osteogenic cell proliferation through protein tyrosine phosphorylation [12,59,61]. A recent study confirmed that the expression of IGF-1, IFG-2, IFG-1R, and IGF-2R increased after NaF treatment, while the expression of TNF-α, IL-1β, RANKL, and cathepsin K decreased. These effects contributed to attenuated alveolar bone resorption caused by a low concentration of NaF [94].

Autophagy is a cellular process that eliminates damaged proteins and organelles, which is important for maintaining cellular homeostasis [95]. Autophagy-related 5 (ATG5) is essential for forming autophagosomes, and ATG5 knockout in mice results in the dysfunction of autophagy and cell death [96]. ATG5 stimulates light chain 3 (LC3)-I and LC3-II and subsequently generates autophagosomes [97]. Beclin-1 helps to recruit ATG proteins and activates downstream genes [98]. Beclin-1 is regarded as an inter-mediator between apoptosis and autophagy, and the dysfunction of autophagy could result in excessive apoptosis [99].

Accumulating evidence suggests that autophagy has an important role in periodontal inflammation [100]. One study demonstrated that a high concentration of NaF induces the apoptosis of human cementoblasts through the inhibition of autophagy. Autophagy-related genes ATG5 and Beclin-1 were suppressed, resulting in alveolar bone resorption [101].

A high concentration of NaF induces oxidative stress, leading to the dysfunction of mitochondria and the stimulation of cell apoptosis [102,103]. Autophagy may serve a protective role in response to fluoride-induced oxidative damage, and the impairment of autophagy results in excessive apoptosis [104,105]. A low concentration of NaF facilitated the expression of osteo-/odontogenic markers of apical papilla cells via the enhanced autophagy pathway [39]. Figure 3 summarizes the biphasic functions of fluoride in periodontal inflammation.

## 6. Fluoride and microRNA

Excessive NaF intake ameliorates dental and skeletal fluorosis. miR-124 and miR-155 are reported to be critical players in fluorosis biology [106]. The recent finding of increased cyclinD1 expression due to miR-486-3p made a significant contribution to understanding the mechanism of skeletal fluorosis [107]. Finding that cyclinD1 is also a direct target of miR-4755-5p, and fluoride exposure suppresses miR-4755-5p and induces cyclinD1 protein in osteoblasts, thus shedding new light upon fluorosis treatment [108]. miR-200c-3p is also considered a potential biomarker for skeletal fluorosis [109]. NaF treatment upregulates miR-200c-3p expression via the BMP4/Smad signaling pathway [109]. NaF exposure mainly affects the Notch, Wnt, hedgehog, and TGF-beta signaling pathways in the osteoblasts [110]. miR-122-5p targets CDK4 protein in NaF-treated human osteoblasts, suggesting its involvement in the etiology of skeletal fluorosis [111]. Skeletal fluorosis may alter the expression level of 10 candidate miRNAs in mouse osteoblast cells through numerous signaling molecules, including autophagy [112]. Female reproductive malfunction could be a result of excessive NaF intake, and the microRNAs (miRNAs) play a substantial role in the regulation of reproduction. A recent study demonstrated that miR-378d was inversely correlated with the autophagy markers under NaF exposure on ovarian cells [113]. Fluoride exposure alters the miR-34 family to orchestrate the downstream signaling molecules in sperm [114]. PIWI-interacting RNAs (piRNAs) are explicitly suggested as candidate biomarkers for fluoride-induced testicular toxicity [115]. Perinatal fluoride exposure can lead to learning and memory problems in mouse offspring, at least in part by the alteration of miR-124 and miR-132 upregulation and their targets [116].

## 7. Clinical Applications

Fluoride use in clinical dentistry dates to the 1920s, with the recognition of its ability to prevent dental caries [117]. An interesting simultaneous development was the observation of the harmful effects of fluoride on dental and skeletal fluorosis [117]. These macroscopic presentations were soon deciphered in the form of microscopic and molecular mechanisms and pathways [117]. The major application of low concentrations of fluoride in dentistry have been related to the prevention of dental caries by fluoridation of systemic water, milk, and salt [118]. There is a plethora of literature on this topic with high-quality evidence supporting the efficacy of this paradigm [117,118]. Fluoride in sodium and stannous and acidulated phosphate forms has found utility in topical applications [119], which have evolved as solutions, gels, varnishes, and pastes [119]. The most recent innovation is the clinical trials evaluating the preventive effects of Silver Diamine Fluoride (SDF) on caries [120]. SDF has a higher concentration of fluoride and a biocompatible association with silver, which arrests the cariogenic process before cavitation and is currently finding global acceptance in pediatric dentistry [120]. Apart from dental caries, the clinical application of a low concentration of fluoride is used in periodontal tissue regeneration [121,122]. A low concentration of fluoride in the form of sodium fluoride is used in root surface biomodification [121,122,123]. The cemental surface gets exposed to bacterial toxins and the inflammatory process in periodontitis [123]. The root surface is modified and allows for the attachment of periodontal fibroblasts [123]. Andreasen et al. evaluated the use of NaF as a root surface biomodification method prior to the replantation of avulsed teeth with an extraoral dry time greater than 60 min [124]. Those results were further evaluated and replicated by several in vitro and animal experiments and attributed to its actions reducing osteoclastic activity [125,126,127]. The role of low concentrations of fluoride has also been explored in the management of external inflammatory root resorption [128,129]. Clinical studies have found that a low concentration of fluoride treatment elevates bone density [130]. In recent years, the adverse clinical effects of fluoride have warranted a search for viable alternatives. The dental remineralization area had witnessed significant changes with the evolution of Casein Phosphopeptide-Amorphous Calcium Phosphate (CCP-ACP) technology. Subsequently, other molecules, such as Novamin, were included in the index, and most recently, the self-assembling peptides have become the cornerstone of contemporary remineralization paradigms [131,132]. Fluoride-free alternatives, including hydroxyapatite, also exhibited beneficial consequences in preventing dental caries, periodontitis, and enamel remineralization and in increasing calcium, phosphorus, and silicon deposition [133,134]. Replacing fluoride with these substances might be a solution to avoid the toxicity risk of high concentrations of fluoride.

## 8. Discussion

The application of NaF in preventing dental caries has been well established. However, both positive and negative effects have been reported. Due to the dual nature of fluoride, it is important to understand the physiology and pathology process during fluoride treatment. Low concentrations of NaF promote wound healing in epithelial cells and BMMSCs. NaF treatment plays crucial roles in maintaining both osteoblasts and osteoclasts, which are important to homeostasis in bone metabolism. An optimal level of NaF could effectively protect against bone resorption. Fluoride consumption affects different types of cells in the body and interferes with various signaling pathways such as inflammation, autophagy, and apoptosis. These advantages of NaF could be used in clinical treatments for periodontal diseases. In contrast, a high concentration of NaF results in fluorosis, oxidative stress, DNA damage, and even toxicity. Thus, finding an appropriate concentration is important for NaF application. Fluoride affects various organs and is implicated in many diseases that have an extensive regulatory network. Exploring the precise mechanisms of the actions of fluoride is crucial for clinical applications. Systematic studies of fluoride treatments will provide more insights into therapeutic applications.

## Figures and Tables

**Figure 1 ijms-23-00962-f001:**
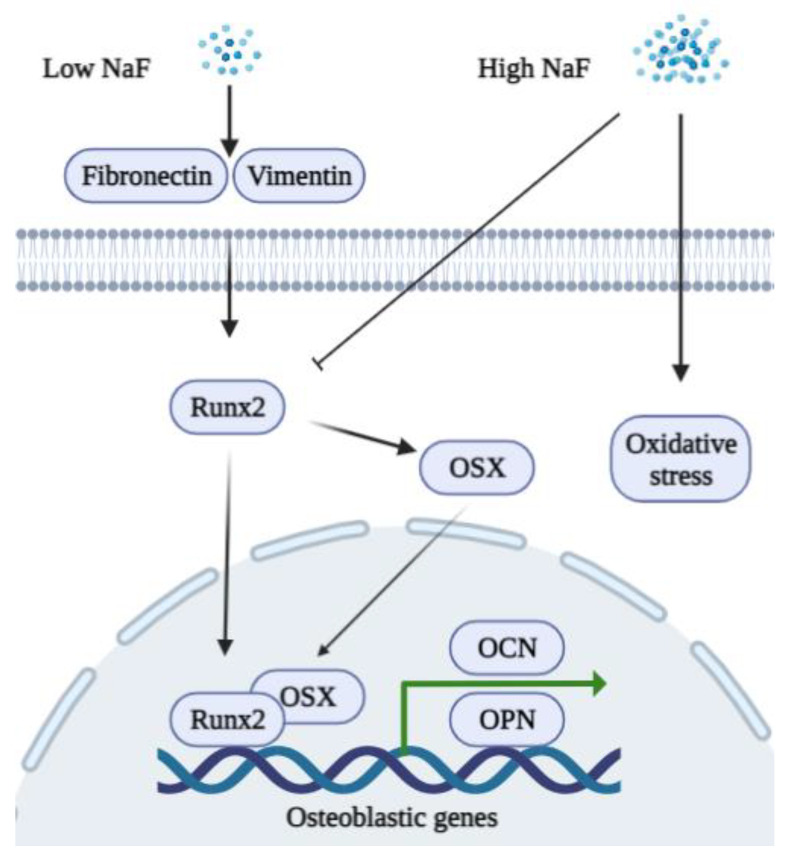
A schematic diagram of the biphasic functions of fluoride in osteoblasts. A low concentration of NaF stimulates the expression of fibronectin, vimentin, Runx2, and OSX to promote the expression of osteoblastic genes (OCN and OPN). A high concentration of NaF inhibits Runx2 and induces oxidative stress in osteoblasts.

**Figure 2 ijms-23-00962-f002:**
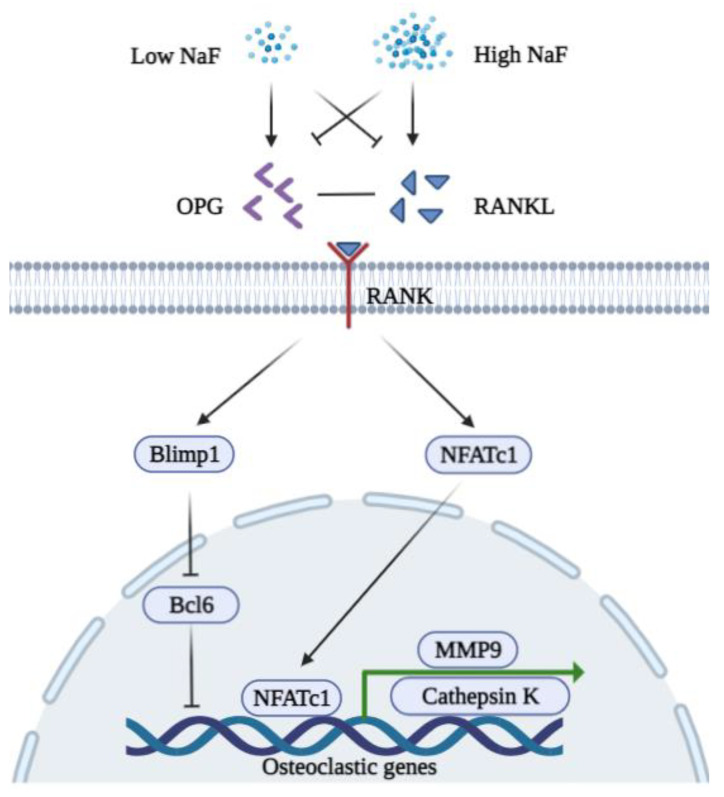
A schematic diagram of the biphasic functions of fluoride in osteoclasts. A low concentration of NaF inhibits the expression of RANKL, NFATc1, MMP9, and Cathepsin K but induces OPG to bind RANKL. A low concentration of NaF is possibly associated with Blimp1/Bcl6 to repress the expression of osteoclastic genes (MMP9 and Cathepsin K). A high concentration of NaF decreases the expression of OPG while inducing RANKL to bind RANK. Induced NFATc1 translocate into the nucleus to promote osteoclastogenesis.

**Figure 3 ijms-23-00962-f003:**
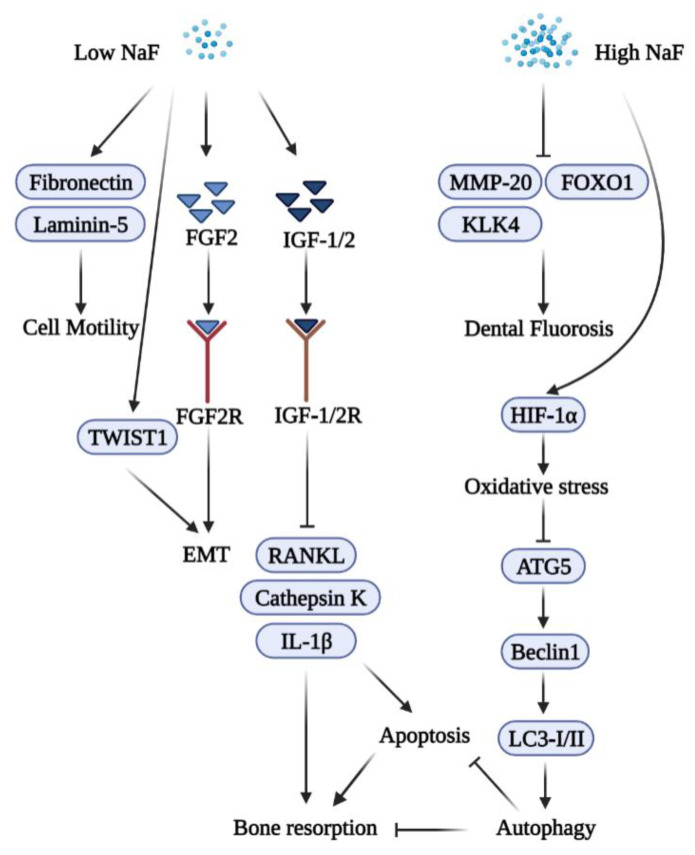
A schematic diagram of the biphasic functions of fluoride in periodontal disease and epithelium cells. A low concentration of NaF induces fibronectin and laminin-5 to improve wound healing, increases the expression of TWIST1 and FGF2 and its receptor to enhance EMT, and attenuates bone resorption via the induction of IGF-1/2 and their receptors which subsequently suppress RANKL, Cathepsin K, and IL-1β. A high concentration of NaF inhibits extracellular matrix proteinases (MMP-20 and KLK4), and FOXO1 results in fluorosis. A high concentration of NaF induces hypoxia-inducible factor 1-alpha (HIF-1α) and oxidative stress, thus inhibits ATG, Beclin, and LC3-I/II to impair autophagy leading to apoptosis and bone resorption.

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
