# Peer review of "Biphasic Functions of Sodium Fluoride (NaF) in Soft and in Hard Periodontal Tissues"

_ijms, 2022, doi:10.3390/ijms23020962_

Round 1

Reviewer 1 Report

This paper reviews the clinical use of fluoride in the form of NaF, with extensive reference to the published literature (132 citations overall).  As such, it is both sound and comprehensive.  The article considers the differences in behaviour of high and low concentrations of fluoride, and shows that the latter are generally beneficial, whereas the former and generally damaging.   The reasons for these differences in terms of differences in cellular and/or metabolic mechanisms are described in detail.   Overall, this is a very useful and timely review, and sums up our current state of knowledge extremely well.

The English is of a very high standard, and I have only two minor suggestions for improvement, as follows:

Line 110: The word "is" should be "are", and the number "4" should be replaced by the word "four".

Reviewer 2 Report

Very  fine paper, well structured. Please check minor spelling and grammar corrections. Well done!

Reviewer 3 Report

Manuscript does not conform to the scientific model of publications and in this form does not bring anything new to current knowledge.
A major revision must be carried out for it to be published.

In my opinion, we need to change the focus of the review and also evaluate the alternative substances to fluoride that can benefit dental and periodontal tissues.
I enclose the reference:

SEM/EDS Evaluation of the Mineral Deposition on a Polymeric Composite Resin of a Toothpaste Containing Biomimetic Zn-Carbonate Hydroxyapatite (microRepair®) in Oral Environment: A Randomized Clinical Trial
Polymers2021, 13(16), 2740

Reformulating the abstract too thin without adding anything new

Incorporate the Pico model into materials and methods and broaden the discussion with alternative approaches to fluoride

Round 2

Reviewer 3 Report

The manuscript has been correctly revised, you can proceed to publication